# Novel and Promising Strategies for Therapy of Post-Transplant Chronic GVHD

**DOI:** 10.3390/ph15091100

**Published:** 2022-09-03

**Authors:** Irina Kostareva, Kirill Kirgizov, Elena Machneva, Nadezhda Ustyuzhanina, Nikolay Nifantiev, Yulia Skvortsova, Irina Shubina, Vera Reshetnikova, Timur Valiev, Svetlana Varfolomeeva, Mikhail Kiselevskiy

**Affiliations:** 1Research Institute of Pediatric Oncology and Hematology of Nikolay Nikolayevich Blokhin National Medical Research Centre of Oncology, Ministry of Health of Russia, 23 Kashirskoe Shosse, 115478 Moscow, Russia; 2Russian Children’s Clinical Hospital of the Nikolay Ivanovich Pirogov Russian National Research Medical University, Ministry of Health of Russia, Leninsky Prospect 117, 117997 Moscow, Russia; 3Laboratory of Glycoconjugate Chemistry, Nikolay Dmitriyevich Zelinsky Institute of Organic Chemistry, Russian Academy of Sciences, Leninsky Prospect 47, 119991 Moscow, Russia; 4Dmitry Rogachev National Medical Research Center of Pediatric Hematology, Oncology and Immunology, Ministry of Health of Russia, 1 Samory Mashela St., 117997 Moscow, Russia; 5Research Institute of Experimental Diagnostics and Therapy of Tumors of N.N. Blokhin National Medical Research Centre of Oncology, Ministry of Health of Russia, 23 Kashirskoe Shosse, 115478 Moscow, Russia; 6Center for Biomedical Engineering, National University of Science and Technology MISIS, Leninsky Prospect 4, 119049 Moscow, Russia

**Keywords:** hematopoietic stem cell transplantation, graft-versus-host disease, novel agents, oligosaccharides

## Abstract

Despite the achievements that have increased viability after the transplantation of allogeneic hematopoietic stem cells (aHSCT), chronic graft-versus-host disease (cGVHD) remains the main cause of late complications and post-transplant deaths. At the moment, therapy alternatives demonstrate limited effectiveness in steroid-refractory illness; in addition, we have no reliable data on the mechanism of this condition. The lack of drugs of choice for the treatment of GVHD underscores the significance of the design of new therapies. Improved understanding of the mechanism of chronic GVHD has secured new therapy goals, and organized diagnostic recommendations and the development of medical tests have ensured a general language and routes for studies in this field. These factors, combined with the rapid development of pharmacology, have helped speed up the search of medicines and medical studies regarding chronic GVHD. At present, we can hope for success in curing this formidable complication. This review summarizes the latest clinical developments in new treatments for chronic GVHD.

## 1. Introduction

Hematopoietic stem cell transplantation (HSCT) is a method of radical treatment of a wide spectrum of malignant and non-malignant diseases, both in children and in adults [1]. One of the most frequent complications of allogenic HSCT is graft-versus-host disease (GVHD) [2]. GVHD is a serious limitation to the success of aHSCT, occurring in donor-derived immune cells in bone marrow or stem cell, where the graft recognizes the transplant recipient as foreign, which results in a negative immune response, starting with tissue damage with subsequently damage to organs and systems, which can lead to death [2,3] (Figure 1). There are two main types of GVHD, called acute graft-versus-host disease (aGVHD) and chronic GVHD (cGVHD), with different clinical manifestations, time of onset, and treatment strategies. Traditional aGVHD, with spotty-papular rash, sickness, emesis, anorexia, profuse diarrhea, enteric obstruction, and cholestatic hepatitis, occurs within 100 days after transplantation.

The system for assessing the severity of chronic GVHD was established at a consensus conference supported by the National Institutes of Health (NIH) in the USA in 2005 and revised in 2014. The GVHD rating system includes data on the number, severity, and areas of affected organs (e.g., skin, mouth, eyes, gastrointestinal tract, liver, lungs, joints/fascia, and reproductive tract) [4]. In 2005, the National Institutes of Health consensus proposed a draft of GVHD criteria, providing an opportunity both to distinguish between acute and chronic GVHD and to diagnose overlapping syndromes. The severity of the organ involvement is rated from 0 to 3, with higher scores reflecting a more severe disease. Based on this information, the overall severity is assessed as mild, moderate, or severe: mild—two or fewer organs/areas without clinically significant functional disorders; moderate—three or more organs/areas without clinically significant operational disorder, or at least one organ/area with clinically significant functional impairment, but without serious disability; and serious—patient without serious disability [5,6].

Although the standardization proposed by the National Institutes of Health Consensus 2005 and 2014 has improved the accuracy of diagnosis and assessment of the cGVHD severity for clinical trials, a number of issues are still to be resolved. Therefore, in 2020, a new attempt was made to design the appropriate tools for recognizing or predicting cGVHD. The presented version of the recommendations particularly discussed bronchiolitis obliterans syndrome (BOS) as the primary diagnostic characteristics of cGVHD, and suggested that the diagnosis of cGVHD could be made on the basis of a single parameter characterizing this condition [7]. Acute GVHD mainly affects the skin, gastrointestinal tract, and liver, whereas cGVHD is not confined by certain organs, but often leads to rather variable clinical symptoms. cGVHD is the most common long-term complication of allogeneic stem cell transplantation (HSCT), developing in more than 50% of patients who had transplantation [8]. The established risk factors for the cGVHD development are previous GVHD, non-compliance of the donor-recipient HLAs, lack of T cell depletion, the elderly age of the recipient, and the use of peripheral blood stem cells [9,10]. Mortality from cGVHD is high, resulting from the dysfunction of the affected organs, infectious complications associated with the delayed recovery of the immune system, or toxicity caused by the prolonged use of immunosuppressive therapy. In particular, these manifestations are clearly observed in severe aGVHD or cGVHD with a progressive onset [11,12].

cGVHD is still considered the most significant cause of late, non-relapse mortality, despite recent interesting results of therapies based on new drugs such as ibrutinib, belumosudil, and ruxolitinib. cGVHD is characterized by an increased production of inflammatory cytokines, activation and proliferation of the alloreactive donor’s T cells, while the immune regulatory mechanisms are unable to balance this pro-inflammatory environment. Patient conditioning regimens, as well as Th1-associated cytokines produced by allogeneic T cells, are the driving factors contributing to the emergence and development of GVHD. Thus, there is an urgent need for the strategies designed to inhibit GVHD pathogenesis by regulating the proliferation of alloreactive donor’s T cells and the production of inflammatory cytokines [13]. cGVHD remains one of the most severe and long-lasting complications of allogenic HSCT and can lead to a complex of urgent problems and disability; it occurs in 30–70% of patients [4]. While major progress has been achieved in understanding the pathophysiology of aGVHD, cGVHD is much less well understood. The pathophysiology of cGVHD differs from that of aGVHD and is primarily described by violations of the mechanisms of immune tolerance influencing inborn and adaptive immunity. Both autoreactive and alloreactive T and B cells play a significant role in the progression of cGVHD [14]. Other pathophysiological agents are the adverse representation of alloantigens by antigen-presenting donor cells and mechanisms of chronic inflammation following the development of cicatrices and fibrosis.

An essential facet of the pathophysiological character of GVHD is the volatility of immune restoration, which depends on age and on the thymus function and hormonal setting [15]. In the beginning, there is tissue damage during conditioning and inflammation with the liberation of pro-inflammatory cytokines—TNF-α, interleukin-6 (IL-6), and interleukin-1 (IL-1). These cytokines, together with the antigens formed as a result of tissue destruction and microbial biocenosis of the intestine, lead to the activation of antigen-presenting cells (ASC). Activated ASCs promote naïve donor T cells and promote the differentiation of helper T cells, cytotoxic T cells (Th1/Tc1—early cytotoxic effectors in the skin and mucous membranes), and T cells that cause tissue damage and the development of fibrosis (Th17/Tc17); they also lead to the expansion of effector T cells that mediate GVHD in other organs, or cGVHD is initiated by naïve T cells that differentiate from pro-inflammatory Th17 cytotoxic T-helpers and follicular T-helpers, with subsequent damage to the thymus and impaired presentation of donor antigens in peripheral tissues. This leads to abnormal activation and differentiation of T- and B-lymphocytes, which together produce cells that secrete antibodies. Antibodies of donor B cells grow in the thymus, spleen, and lymph nodes, ameliorating the infiltration of Th17 in the skin. The blocking of autoantibody-mediated signaling tracts in cGVHD receivers may yet present new ways to ameliorate cGVHD [16].

## 2. Approaches to cGVHD Diagnosis and Treatment

cGVHD typically starts between 3 months and 2 years after HSCT. Besides classical manifestations such as poikiloderma, sclerotic changes, and sclerosed lichenoids, cGVHD can mimic almost every autoimmune disease. Since cGVHD can affect a number of organs, and patients often do not report changes until functional disorders are recognized, regular examination of all potentially affected organs is necessary. As predictors of the progression and severity of chronic GVHD, HSCT from female donors to male recipients and previous episodes of acute stage II–IV GVHD are noted. The group of severe cGVHD is associated with low overall survival and higher mortality [17].

The goal of treating chronic GVHD, like for any chronic disease, is to improve the patient’s condition, reduce the severity of symptoms, control disease activity, prevent the development of irreversible changes and disability, and minimize the toxic effects of therapy [18]. A long-term task is the creation of immuno-tolerance, which allows for canceling without the risk of recurrence of symptoms. First-line care includes steroids administered separately or in combination with calcineurin inhibitors (CNI) [19].

As described above, prednisone (1 mg/kg per day) (Figure 2) is our first drug of choice for patients who require systemic therapy for chronic GVHD. Additional therapy is necessary for patients with progression after two weeks of taking prednisone or no response after four to six weeks [20]. The ideal therapy in this situation is unknown, and patients should be encouraged to participate in clinical trials. For those who cannot or do not want to participate in studies, we suggest adding a CNI (for example, cyclosporine or tacrolimus; Figure 2) [21]. Cyclosporine and tacrolimus share similar mechanisms of effect, anticipated medical effectiveness, and toxic effects, including nephrotoxicity, hyperkalemia, hypertension, and hypomagnesemia. Significant side effects include thrombotic microangiopathy associated with the transplant and neurotoxic effects, which can lead to early termination of administration. It is important to remember that cyclosporine and tacrolimus are nephrotoxic. As a result, the use of other nephrotoxic drugs should be avoided if possible, so that the agent can be delivered in targeted doses [22]. Patients should stop taking medications that have proved ineffective. Sirolimus (Figure 2) can also provide benefits over other immunosuppressants. In pilot simulations, sirolimus contributes to the spread of normative T cells, assuming that it can retain the effects of the graft against the tumor and defend against chronic GVHD [23]. As a rule, drugs that have shown their ineffectiveness should be gradually reduced, but no more than one medication should be changed at a time so that their efficacy can be evaluated [22,23].

Ruxolitinib is an oral JAK1/2 kinase inhibitor of the signaling pathways that play an important role in the activation of the immune cells and tissue inflammation in patients with GVHD. In 2019, ruxolitinib was approved by Food and Drug Administration (FDA) as a treatment for steroid-refractory aGVHD (SR-cGVHD) [24]. The results of phase III trial of patients with SR-cGVHD showed a higher ruxolitinib effectiveness as compared to current therapies. Hemocytopenia was the most frequent adverse event observed in patients with GVHD in that study, which was consistent with the previously published data. Moreover, Moiseev et al. [25] reported that the severity of neutropenia and thrombocytopenia was associated with cytomegalovirus (CMV) reactivation (*p* = 0.07), treatment with ganciclovir (*p* = 0.0006), and a higher initial dose of steroids (*p* = 0.0017). Ruxolitinib is now approved by the FDA for cGVHD [26].

## 3. Cell Technologies and Extracorporeal Methods

### 3.1. Extracorporeal Photopheresis

The best treatment for patients with advanced or resistant disease is unknown, and clinical practice varies. The main treatment options are non-pharmacological methods of treatment, such as extracorporeal photopheresis (ECP) and ultraviolet irradiation therapy with psoralen, as well as the use of additional immunosuppressive drugs or cellular technologies. When choosing from among these agents, it is necessary to take into account the organs involved, the patient’s concomitant diseases, the doctor’s experience, and available resources. In this situation, we often consider the use of extracorporeal photopheresis in addition to prednisone and a calcineurin inhibitor, since it provides a relatively high response rate without the addition of drugs with potential side effects [27].

The ECP is a therapy that involves leukapheresis. The lymphocyte-enriched patient’s leukocyte suspension was exposed to extracorporeal treatment with a photosensitizer 8-methoxypsoralene (8-MOP) with the following ultraviolet irradiation and reinfused to the patient. The scheme most often described in the above-mentioned studies follows a pre-determined schedule with two procedures per week for 4 weeks, followed by a reduction to two procedures every second week for the next 2 months, and then two procedures per month [28]. Nordic ECP Quality Group analyzed 26 studies for ECP use in acute GVHD and 36 in chronic GVHD [29]. The results showed that most patients who underwent ECP had a partial response or even better outcomes, though the response rate was determined by the affected organs. However, the quality of evidence was regarded as low–moderate since the main reports on the effectiveness of ECP in patients with cGVHD were based on small, uncontrolled, retrospective studies with different endpoints and treatment regimens (GRADE).

The first prospective randomized controlled clinical study of the effectiveness of TEC in the treatment of chronic GVHD used an intensive treatment regimen with weekly TEC therapy for 12 weeks, whereas most other studies used weekly treatment for 4 weeks or the start of treatment every two weeks. The authors evaluated the effectiveness of ECP by the Total Skin Score (TSS) using a validated ordinal 50-point whole body scoring system. Most patients of the ECP group achieved a reduction in steroid drug doses by at least 50% over the 12-week study period, though no statistical difference in TSS decrease was found between patients of the ECP group and the control one. The authors concluded that the low efficiency of ECP was the result of a short period of the ECP course [30]. The duration of the ECP course was prolonged up to 24 weeks in a later randomized trial. The results of the study with these conditions showed an improvement in cutaneous and extra-cutaneous GVHD in patients who previously had cGVHD aggravation or no clinical effectiveness when receiving only standard immunosuppressive therapy [31].

A recent randomized controlled trial evaluated the effectiveness of the first-line standard therapy in combination with ECP in patients with moderate or severe cGVHD by NIH diagnostic and response criteria. The authors found an improvement in the quality of life of patients of the ECP group, although there were no significant differences in effectiveness compared to the control group [32]. Clinical studies have reported the safety of the ECP procedure; however, in 2018, the FDA registered seven cases of pulmonary embolism during or shortly after the therapy, including four patients undergoing treatment for GVHD that resulted in two deaths. In addition, two patients had deep vein thrombosis of the limb. The authors suggested that ECP therapy may increase the risk of thrombosis since, generally, the patients with GVHD are at increased risk of thrombosis [33,34]. ECP therapy in patients with GVHD was effective due to the modulation of various subpopulations of T-helpers; in particular, a reduction in Th22. Increased expression of FasR on CD4^+^ CD8^+^ T cells seemed to be the result of elevated Fas-initiated pro-apoptotic signaling [35,36]. Increased circulating CD4^+^/CD25^+^/FoxP3^+^T-regulatory cells (Treg) were observed after the ECP procedure [37,38]. The experimental studies showed that Treg increase was determined by the enhanced IL-10 production after the ECP [39]. Treg proliferation and functioning is mostly stimulated by IL-2 [40].

A phase II study aimed to increase the effectiveness of ECP in patients with SR-cGVHD was performed assessing ECP with subcutaneous administration of low doses (1 × 10^6^ IU/m^2^) of IL-2 (LD IL-2) for the following 8 weeks. Although this study did not provide a definitive conclusion whether the combined treatment of ECP plus IL-2 was more beneficial or equivalent to each monotherapy, it was noted that the 16-week course of treatment was safe and the objective clinical response reached 62% of adult patients with advanced SR-cGVHD. The patients with a clinical effect tended to have a greater increase in Treg content compared to that of the patients who had no response during the 16-week study [41].

### 3.2. Interleukine-2 and T Regulatory Cells

A phase II clinical trial evaluated the efficacy of LD IL-2 (1 × 10^6^ IU/m^2^) monotherapy in patients with SR-cGVHD. IL-2 was administered subcutaneously for 12 weeks; the second course was started no earlier than 4 weeks later. LD IL-2 therapy induced rapid selective expansion of Tregs and NK cells in all patients treated. The increased number of peripheral Tregs peaked after 4 weeks, though continuing therapy did not lead to a further increase. No exacerbations of cGVHD were registered after the start of IL-2 therapy. An objective partial response was observed in 20 out of 33 patients (61%), involving several cGVHD localizations [42]. A recent analysis of five phase I/II clinical trials evaluating the LD IL-2 efficacy included 105 adult patients. The objective response rate after 8 or 12 weeks of LD IL-2 accounted for 48.6% and 53.3%, respectively. LD IL-2 therapy resulted in rapid expansion of peripheral CD4 Tregs in all patients without significant alterations in CD4^+^T- or CD8^+^T cells. LD IL-2 therapy may be more effective at the early stages of cGVHD or in combination with corticosteroids as the first-line therapy before the development of extensive fibrosis and tissue destruction. Besides, the authors suggested that a higher effectiveness of LD IL-2 therapy could be achieved as a result of a combination therapy with ibrutinib, ruxolitinib, or belumosudil, which specifically inhibited the signaling pathways in B cells or effector T cells [43].

Given Treg’s clinical significance for therapy, it seemed important to increase the number of Tregs by extracorporeal expansion and adoptive transfer to patients with cGVHD. Preclinical studies have shown that adoptive Treg transfer can prevent allograft rejection and reduce the severity of GVHD by inhibiting alloimmune reactions [44]. Tregs represent a minor population of human lymphocytes (2–4%); therefore, extracorporeal expansion in the presence of IL-2 of this suppressor lymphocyte subpopulation is necessary for their use in clinical practice. Donor’s peripheral blood lymphocytes, mostly of the same donor whose hematopoietic stem cells were used for allo-HSCT, or umbilical cord blood (UCB) are used as a source of Tregs [45]. UCB is an important source since it includes a high Treg content and low number of memory T cells compared to the peripheral blood of an adult [46]. However, adoptive Treg therapy of cGVHD has been studied only in a few small clinical and pilot trials. One of the first clinical trials of the safety and efficacy of Tregs from umbilical cord blood with doses of 1–30 × 10^5^ Treg/kg included twenty-three patients. The results demonstrated the safety of administration of UCB Tregs at a dose of 30 × 105 cell/kg and a reduction in the risk of developing cGVHD as compared to the retrospective control [44]. Later, the authors evaluated the preventive effect of higher doses of Tregs (3 to 100 × 106 cells/kg) in eleven patients who received the same conditioning regimen to suppress immunity with sirolimus and mycophenolate mofetil. The median ratio of CD4 ^+^ FoxP3^+^ CD127^–^ in the infusion suspension was 87% (78–95%); no dose-limiting adverse events were observed for this infusion regimen. The rate of grade II–IV cGVHD accounted for 9% compared to 45% in the control group. One year later cGVHD was 0% in the Tregs group and 14% in the control. Hematopoiesis recovery and chimerism, cumulative infection density, and relapse-free survival were similar in Treg recipients and the control group.

A phase I study included 25 patients with SR-cGVHD who received donor Tregs in the doses of 0.3–1.0 cells/kg in combination with LD IL-2. The results showed that 5 of 25 patients (20%) with SR-cGVHD had partial responses (PR). Eighteen patients (72%) had disease stabilization (SD), including 10 patients with minimal response (MR) that did not meet the PR criteria [47].

Three children with severe SR-cGVHD underwent adoptive immunotherapy with Tregs from a stem cell donor. Supportive immunosuppression of the third line included cyclosporine A and low doses of steroids. Patients showed marked clinical improvements with a significant decrease in GvHD activity. Laboratory tests revealed a significant improvement of the immunological graft engraftment, including lymphocytes and dendritic cells [48]. These data indicate the feasibility of this approach, though more extensive clinical trials are required to assess the safety and effectiveness of adoptive Treg therapy to cut the active cGVHD symptoms.

Recently, genetically modified CAR-Tregs have been proposed for GVHD suppression. Donor HLA molecules are interesting candidate target-antigens for CAR-Treg transplantation. The studies on the experimental showed that the adoptive transfer of hA2-CAR Tregs suppressed HLA-A2^+^ cell-mediated xenogenic “graft-versus-host” reaction in mice [49]. Bw6 is an MHC epitope identified as a frequent graft alloantigen due to its incorporation into human leukocyte antigen molecules. An engineered Bw6-specific CAR-Treg product caused a suppressive response to a specific antigen without a cytotoxic or pro-inflammatory effect in vitro. Bw6-CAR-Tregs in the dose of 6.54 × 10^6^ cells/kg caused no obvious signs of toxicity and were found mainly in the recipient’s bone marrow and persisted for at least 1 month. Considering the experimental data, the authors made a conclusion about the safety of CAR-Tregs in organ transplantation [50].

### 3.3. Mesenchymal Stem Cells

The potential of mesenchymal stem cells (MSCs) to alter the immune response and immediate hematopoiesis has been discussed in terms of MSC use for the treatment of complications of allogeneic hematopoietic stem cell transplantation [51].

MSC immunosuppressive activity is determined by the production of cytokines and growth factors [52]. Studies on animal models have shown that MSCs stimulate the engraftment of hematopoietic stem cells (HSCs) and prevent transplant rejection. A number of clinical studies have confirmed the effectiveness of intravenous infusion of MSCs for GVHD prevention or treatment in patients after HSC transplantation. The first report demonstrating the clinical efficacy of GVHD treatment with MSCS infusion was published in 2004 [53]. Since then, numerous clinical trials of this technology have been performed, reporting controversial data. In 2013, Muroi et al. published the results of a multicenter phase I/II study of MSCs from the bone marrow of healthy unrelated volunteers for the treatment of steroid-resistant acute GVHD. The study included 14 patients with hematological malignancies and GVHD grade II/III. MSCs were administered intravenously at a dose of 2 × 10^6^ cells/kg 2 times a week for 4 weeks. The clinical effect of MSC therapy was registered in 13 out of 14 patients (92.9%) by the end of the course. No marked side effects associated with the infusion of MSCs were observed in the study [54]. Later, these authors presented the results of a phase II/III clinical study of the efficacy of MSC therapy in patients with steroid-refractory GVHD. The study included 25 patients with GVHD grade III/IV. In total, 36% of patients reported a complete or partial response at the end of the 4-week course of MSC therapy. The survival rate of patients with a clinical response to MSC therapy was significantly higher than in patients without a response. Side effects associated with MSC infusions were not observed. Taking into account the results of these two clinical studies, the authors made a conclusion about the effectiveness of MSC for the treatment of steroid-resistant GVHD [55].

However, an Australian clinical trial evaluating the effectiveness of MSCs in 87 children with allo-HSC reported that MSC treatment showed neither an improvement in survival nor any significant clinical effect [56]. In addition, the results of the later studies demonstrated no significant improvement in the survival of patients with GVHD after MSC therapy [57,58]. However, it should be noted that MSCs were received from different volunteers, and the MSC cell products were generated ex vivo under different cultivation conditions, which significantly affected the MSC activity.

The combination of MSCs with basiliximab, a calcineurin inhibitor, evaluated in an open randomized trial phase III showed that the combination in patients with SR-GVHD led to a significant improvement in the therapy efficacy, with a higher overall response (OR) rate on Day 28 and Day 56 than that in the control group. Moreover, the authors found that MSCs could reduce the severity of the side effects of second-line therapies, in particular, BM toxicity and infections [59].

One of the possible ways to increase the effectiveness of MSCs in cGVHD treatment suggests the injection of MSCs directly into the bone marrow. According to clinical studies, such therapeutic approach can improve the engraftment of transplanted HSCs compared with hematopoietic stem cell intravenous injection. Although the potential of MSCs for cGVHD therapy seems undoubted, it is necessary to take into account the profile of pro-inflammatory/anti-inflammatory cytokine production, which is, to a great extent, determined by the conditions of MSC cultivation, as well as the way of administration and the dosage of MSCs [53,60,61].

## 4. Novel Agents in cGVHD Treatment

While first-line treatment is based on randomized studies, second-line therapy is mostly based on stage II trials, and a retrospective review is accessible. Some medications, such as imatinib and retinoids, are recommended solely for occurrences related to sclerosing obliterating bronchiolitis (imatinib; Figure 2) or sclerodermoid dermal alterations (retinoids, imatinib; Figure 2) due to their specific mechanisms of effect. Many drugs that are widely applied for the therapy of acute and chronic leukemias, myeloma, and lymphoma and are used in an attempt to treat cGVHD (ibrutinib, ixazomib, bortezomib, imatinib; Figure 2). Further, the issues of tolerability, toxicity, and efficacy in patients who have undergone allotransplantation and received various immunosuppressive drugs are still not fully understood. In adult patients, ibrutinib has proven to be effective and, according to clinical studies, is well tolerated [62]. However, in addition, there are a number of drugs, such as JAK 1/2 inhibitors, which are at the late stages of studies and are already used as a treatment for the effects of GVHD, such as those in the liver, intestines, and skin. The kinases JAK1 and JAK2—associated with the cytokine receptor—are critical for the inflammatory cytokine response in GVHD. 

### 4.1. Selective Inhibitor of Janus Kinases

In many studies, ruxolitinib, a selective inhibitor of Janus kinases (JAK1 and JAK2), has shown possible efficiency in sufferers of acute GVHD that is refractory to glucocorticoids [63]. Zeiser et al., in 2020, performed a multicenter randomized open phase 3 study in which the effectiveness and reliability of oral administration of ruxolitinib were compared with those of the treatment chosen by the investigator in sufferers with an acute response to graft-versus-host glucocorticoids (GVHD) following allogeneic stem cell transplantation. The initial terminus of this trial was the total response on Day 28, and randomization was allocated according to the primary grade of acute GVHD (II, III, and IV) [64]. Ibrutinib, a Bruton tyrosine kinase inhibitor (BTK), which is used to treat certain lymphoid malignancies (for example, mantle cell lymphoma and chronic lymphocytic leukemia marginal zone lymphoma), has activity against chronic GVHD [65]. Ibrutinib is approved in the USA for the therapy of chronic GVHD. 

### 4.2. Selective Inhibitor of Tyrosine Kinase

Ibrutinib is a selective and irreversible inhibitor of the Bruton’s tyrosine kinase. FDA-approval of ibrutinib as second-line therapy of steroid-refractory or steroid-resistant cGVHD. In the period from 2014 to 2017, a group of scientists from Stanford University conducted a study that included patients over the age of 18 with steroid-dependent or refractory GVHD after hematopoietic stem cell transplantation. The steroid-dependent disease was identified as cGVHD, requiring the administration of prednisone at 0.25 mg/kg/day for 12 weeks; resistant illness was defined as advanced cGVHD, regardless of treatment with prednisone at 0.5 mg/kg/day for 4 weeks. Active cGVHD was required, and patients were selected on the basis of both an erythematous eruption on the skin of the body of >25% or a National Institutes of Health (NIH) oral cavity index of >4 [66]. These manifestations were chosen because it was expected that they would respond quickly to effective therapy and, consequently, the patients could potentially avoid the long-term effects of inefficient therapy. Therapy was started with a dose of 420 mg of ibrutinib, while 6 to 27 patients were analyzed in Stage 1b, depending on the rate of dose-limiting toxicity (DLT) and the necessity to downsize the dose. The patients underwent either myeloablative or non-myeloablative stem cell transplantation for various malignancies. The oral cavity and skin were the most frequently affected organs, and 85% of patients showed signs of cGVHD in two organs [67]. Of 42 patients, 28 had steroid-dependent GVHD, 6 had steroid-resistant GVHD, and 8 had steroid-dependent and -resistant forms of the illness in their history. With a mean follow-up period of 13.9 months, 12 patients (29%) continued to receive ibrutinib, and 30 (71%) discontinued treatment. The length of therapy was from 5.6 to 24.9 months for the 12 patients who went on treatment. The most frequent causes of termination of treatment were NYA (*n* = 14), progression of cGVHD (*n* = 5), or the patient’s decision (*n* = 6); two patients stopped treatment following the extinction of symptoms of GVHD [68].

### 4.3. Ceramide

Ceramide is a key structural element of glycosphingolipids involved in a number of molecular signaling pathways and cell regulation [69]. Ceramide is embedded in the cell membranes and plays a central role in the biological activity of glycosphingolipids. Ceramide biosynthesis proceeds with the participation of ceramide synthases (CerS 1–6). The experimental studies on a mouse model showed that the inhibitor CerS6 ST1072 prevented and decreased cGVHD severity. Specific CerS6 inhibition involved in TCR signaling suppressed the migration of donor T cells to GVHD target organs [70].

### 4.4. Selective Inhibitor of Rho-Associated Kinase

Rho-associated coiled-coil-containing protein kinase (ROCK) is a signaling pathway that modulates the inflammatory response and fibrous processes and is disrupted in autoimmune diseases. Rho-associated kinase 2 (ROCK2) is involved in the regulation of interleukin-21 (IL-21) secretion, which plays a major role in autoimmunity [71]. Belumosudil and bendamustine are selective oral inhibitors of ROCK 2, which have demonstrated safety and efficacy in cGVHD treatment. The research showed that belumosudil (BEN) was effective in combination with cyclophosphamide (CY) for the prevention of the graft-versus-host reaction in the course of T-cell haploidentical bone marrow transplantation. Partial replacement of CY with bendamustine (BEN) was well tolerated and led to earlier graft engraftment. Post-transplantation therapy with CY+BEN resulted in a significant decrease in cGVHD cases and increased relapse-free survival as compared to that of PT-CY alone [72,73]. Belumosudil reduces Th17 and follicular helper cells via downregulation of STAT3 and enhances regulatory T cells via upregulation of STAT5. Preclinical studies have shown that belumosudil reduces fibrosis by inhibiting fibroblast differentiation and collagen production. [74]. Clinical studies of belumosudil in patients with cGVHD have demonstrated a significant improvement in the quality of life of these patients. In particular, belumosudil at a dose of 200 mg/day and 200 mg twice a day, in patients with cGVHD who had previously received from 2 to 5 lines of therapy, led to a decrease in the grade of symptoms in 59% and 62% of patients, respectively [75]. Based on the results of clinical studies in 2021, the US FDA approved belumosudil for the treatment of adults and children aged 12 years and older with cGVHD after the ineffectiveness of at least two previous lines of systemic therapy [76]. Yalniz et al. [77] analyzed the cost-effectiveness of the frequently utilized agents in steroid-refractory cGVHD, including tacrolimus, sirolimus, rituximab, ruxolitinib, hydroxychloroquine, imatinib, bortezomib, ibrutinib, extracorporeal photopheresis, pomalidomide, and methotrexate. The authors found pomalidomide to be the least cost-effective treatment for eyes, gastrointestinal, fascia/joints, skin, and oral GVHD, and imatinib was found to be the least cost-effective treatment for liver and extracorporeal photopheresis in pulmonary GVHD (Figure 2). However, the most optimal strategy in terms of cost-effectiveness was methotrexate for all of the organ systems.

## 5. Experimental Therapy

New therapeutic approaches are aimed at suppressing the terminal stages of Th17/Tfh (follicular T helper) cell development using small RORyt molecules, inhibiting interleukin-17/21, inhibiting kinases, inhibiting STAT3, restoring regulatory T cells, and inhibiting CSF-1 (colony-stimulating factor 1), protecting the thymus. Currently, low doses of IL-2 in vivo are actively used for the expansion of regulatory T cells. Therapy aimed at damaged B cells includes anti-CD20 monoclonal antibodies, which reduce the severity of manifestations of chronic GVHD; however, this is more effective with preventive administration, before the formation of actively antibody-producing plasmablasts and plasmacytes [78]. Thus, IL-2 can cause reactions in chronic steroid-resistant GVHD, but further studies must still be completed before including it in standard regimens. Interleukin-2 (IL-2) is a cytokine derived from T cells that plays a key role in immune reactions, as well as in the development and function of regulatory T cells [79]. Animal studies have shown that Treg cells can control GVHD and that IL-2 depletion due to calcineurin inhibition reduces Treg activity. In 2016, scientists published a phase 2 study on the use of IL-2 (at a dose 1 × 10^6^ international units/m^2^) in 35 adults with glucocorticoid-resistant chronic GVHD; clinical responses were reported in 20 out of 33 assessable patients (not completed) by Week 12; in 10 patients, the disease was stable, and in 3, it was progressive. Adverse events resulted in a dose decrease in five patients and premature discontinuation of therapy in two [80].

The search for new agents continues, and patients should be encouraged to participate in clinical trials. It should be remembered that differences in the overall response rate may be related to the characteristics of the included patients, such as the presence or absence of signs of high risk, the time of occurrence of GVHD, and previous treatment [81].

Among the drugs that are currently undergoing preclinical studies, we can distinguish etanercept, thalidomide (studies have shown that the overall response to treatment ranges from 20 to 65%), and pentostatin. The latter drug is the most studied. In 2017, researchers issued a phase II trial of pentostatin in patients with a chronic graft-versus-host reaction resistant to corticosteroids. Fifty-eight patients were registered who had received intensive preliminary treatment (median, four previous regimens; average age, 33 years). An initial drug dose of 4 mg/m^2^ intravenously every two weeks for 12 doses, continuing until a benefit was documented, was prescribed to patients with cGVHD who had previously received intensive treatment (on average, four previous regimens). An objective response was observed in 32 out of 58 patients (55%), and the overall one-year survival rate was 78%. A similar response rate was observed in a phase II study of this drug in 51 children with chronic GVHD resistant to corticosteroids [82,83].

## 6. Promising New Agents for Treatment of Hematopoietic Dysfunction and Hemostatic Disorders in cGVHD

Disorders of hemostasis and hematopoiesis in patients with cGVHD are significant complications of a multifactor etiology, including tissue damage with the release of microparticles, cytokine release, clearance of macrophages/monocytes, CMV infection, the production of TNF-β, and decreased thrombopoietin level [84].

The restoration of the hematopoietic and immune systems after HSCT is a complex process that can take from several months to several years [85]. The HSC recipients after conditioning have an aplastic period characterized by severe neutropenia, anemia, and thrombocytopenia, which leads to a high risk of infections and hemorrhagic complications. Timely recovery of donor’s and self-immune cells is of paramount importance for preventing HSCT-associated complications and is one of the main predictors of HSCT outcomes [86]. Immunosuppressive therapy, including cyclophosphamide, in patients with steroid refractory GVHD, as well as modern targeted drugs, inhibitors of pro-inflammatory cytokines and their receptors, lead to hematopoiesis depression [87].

Patients with cGVHD receiving allogeneic HSCT often develop hemostatic disorders that are represented by intravenous thromboembolism (ITE) and bleeding. A cohort study involving 1514 recipients with allogeneic and autologous HSCT showed an ITE in 4.6% of patients. A later clinical study, which analyzed 2276 patients who underwent allo-HSCT for 4 years, demonstrated a high ITE incidence, reaching 8.3%. Moreover, acute and chronic GVHD were independent risk factors for ITE development [88]. A retrospective cohort study of BMTSS evaluated cases of ITE in patients after allo-HSCT who lived at least 2 years after transplantation. The control group consisted of brothers and sisters of patients without signs of malignant neoplasms. The authors found that the probability of ITE in patients after allo-HSCT was 7.3 times higher than in the control group. Obviously, immune dysregulation and the use of immune-suppressive drugs in the treatment of GVHD could lead to an increased risk of ITE [89]. It should be noted that bleeding is also a significant complication of allo-HSCT and can be fatal [90].

Generally, prevention of ITE implies anticoagulant therapy, including low-molecular-weight heparin [88]. The most significant undesirable phenomenon of anticoagulant therapy is heparin-induced thrombocytopenia (HIT), an immuno-mediated complication of unfractionated heparin (UFH) or low-molecular-weight heparin (LMWH), leading to transient thrombocytopenia and accompanied by a prothrombotic condition [91,92,93]. Given the susceptibility of patients with cGVHD to thrombocytopenia, prolonged use of UFH and LMWH may lead to an increase in the frequency of HIT and cause the development of bleeding. Thus, new compounds without undesirable effects are required for the treatment and prevention of hematopoietic dysfunction and hemostatic disorders in cGVHD.

One of the promising classes of substances with hemostimulating, antithrombotic, and anticoagulant activity is a natural sulfated polysaccharide, fucoidans (Figure 3), derived from brown algae in such amounts it is sufficient for drug production. Fucoidans and their modified derivatives, such as heparin, have anticoagulant and fibrinolytic activity, which makes them an essential base of potential compounds for the prevention of thrombosis and thrombolysis [94,95,96].

The mechanisms of anticoagulant and antithrombotic activity of fucoidans include effects on both the external and internal coagulation pathways [97,98]. Despite the similarity of the effects, the mechanism of the antithrombotic action of fucoidans is different from that of heparin and has a certain impact on the processes at the final stage of coagulation—the conversion of fibrinogen into fibrin under the influence of thrombin. An advantage of fucoidans with an antithrombotic effect for venous and arterial thrombosis is a low hemorrhagic risk [99]. This specific feature of fucoidans may be important for long-term anticoagulant therapy in GVHD patients prone to hemorrhagic reactions.

Another essential characteristic of fucoidans is their hemostimulating activity. Excessive immune activation in patients with GVHD damages both hematopoietic stem cells and progenitor cells, as well as the surrounding bone marrow niche, which leads to the development of cytopenia, represented mainly by a decreased number of neutrophils and platelets. Hematopoietic damage may be aggravated by emerging HCT-related complications and their treatment with immune suppressants, including cyclophosphamide [100]. Similar to G-CSF, various polysaccharide compounds can stimulate hematopoiesis [101,102]. On the other hand, experimental models have shown that, unlike G-CSF, fucoidan derivatives neutralize neutropenia and also contribute to the restoration of lymphocyte, erythrocyte, and platelet numbers, causing the mobilization of progenitor cells [103,104,105].

Similar biological activity was shown for another type of sulfated polysaccharides—fucosylated chondroitin sulfates. Recent studies have demonstrated that fucosylated chondroitin sulfates exhibit larger pharmaceutical potential in connection to GVHD because of their ability to stimulate the release of white and red blood cells, as well as platelets from bone marrow in cyclophosphamide-treated mice [101].

A critical factor for pharmaceutical use of fucoidans is their complex and irregular structures, which is challenge for manufacturing according to GMP standards. Different laboratories thoroughly investigate alternative biopolymers with more regular structures. In this context, the fucosylated chondroitin sulfates are especially interesting [105,106,107,108,109]. The structure of these polysaccharides is also complex (for examples, see Figure 4 and the literature cited), but more regular compared with the structure of fucoidans. Nevertheless, some fucosylated chondroitin sulfates can represent more complex mixtures, in particular the polysaccharides from *Cucumaria japonica* [110] and *Cucumaria frondosa* [111], which includes an additional O-sulfated fucosyl-unit as R^3^ (depicted in Figure 4).

The stereoselective chemical synthesis of appropriate oligosaccharides, which represent pharmacophore fragments of fucoidans and fucosylated chondroitin sulfates, opens a good way forward for developing innovative therapeutics suitable for application in GVHD. Modern synthetic methods permit the preparation of such a compound of rather big size and complex structure [112,113,114].

## 7. Conclusions

The last decade has led to significant improvements in allogeneic stem cell transplantation, with only one notable exclusion: the treatment of patients with GVHD has not changed substantially since a clinical study in which corticosteroids were defined as the norm for initial treatment [115]. Developing a consensus on cGVHD treatment standards is the main obstacle to the success of transplantation. Further, a more complete understanding of the pathogenesis, risk factors, and choice of cGVHD therapy deserves increased attention. Chronic GVHD is an immunologic assault of host organs or fabrics by donor T and B cells after HSCT [116]. Donor T-helper cells (Th) play a crucial role in the triggering of GVHD due to their capacity to distinguish into Th1 (secreting IL-2 and IFN-γ), Th2 (secreting IL-4, IL-5, IL-10, and IL-13), Th17 (secreting IL-17), and Tfh cells, contributing to organ-specific GVHD [117]. Many studies have revealed that cGVHD is a highly selective activation of alloreactive donor CD4^+^ T cells that are called to help host B cells, thereby inducing B cell activation and autoantibody production [3].

Current treatment options for GVHD mainly involve the use of corticosteroids. Due to their lymphopenic and anti-inflammatory properties, and generally based on controlled medical studies, corticosteroids have remained the “gold standard” of first-line therapy for the treatment of cGVHD [ 78]. Corticosteroids may be given alone or in combination with other immunosuppressive drugs, such as calcineurin inhibitors, which suppress the immune system by preventing T cells from making IL-2. However, this method of treatment can lead to an increase in the recurrence of malignant hematopoietic disease [118]. It should be noted that long-term use of corticosteroids carries a number of complications, such as infections, hyperglycemia, decreased bone mass, and avascular necrosis. We could say that the treatment of GVHD has three separate purposes: reducing the activated status of B and T cells (for example, JAK1/2 inhibitors can reduce the activity of Th1, Th2, and Th17, and monoclonal antibodies (mAb) act against B and T cells), exerting a pro-inflammatory effect (reducing the secretion of IL-6, TNF-α, and IL-17—for example, infliximab and etanercept), and slowing the development of fibrosis (CSF inhibitors-1, pathways of TGF-β, PDGF, spleen tyrosine kinase (Syk), and Rho-associated kinase 2) [119].

A summary of the recent advances in preclinical and clinical studies involving some new treatment strategies for cGVHD based on immunotherapy that include monoclonal anti-B and T cell antibodies, as well as TKI, JAK, MEK, proteasomes, and PI3K inhibitors as potential treatment options, have revealed second and long-range lines for cGVHD [120]. Each of these new strategies opens up great opportunities for preventing or reducing cGVHD, since they are aimed at restoring/maintaining immune regulation by affecting B cells (responsible for the production of autoreactive antibodies) or T lymphocytes (responsible for the production of pro-inflammatory and profibrotic cytokines), as well as by reducing inflammation, which is essential in the pathogenesis of cGVHD [121]. However, the largest disadvantage of some of these new methods of treating cGVHD is the information about their tolerability and effectiveness obtained from preclinical or clinical trials [122]. In addition, there are a number of drugs, such as TI and JAK 1/2 inhibitors, which are in the late stages of studies and are now being exploited as a last-resort treatment [123].

Survivors of HSCT are at risk of developing treatment-related sequelae that may manifest years after treatment. These complications have become a serious cause of increased mortality, and it is recommended to screen some of these conditions in the hope that early detection can lead to more effective treatment. Patients with more extensive (moderate to severe) GVHD, especially with multiple organ pathology, have unfavorable long-term results [124]. The complexity of the treatment of cGVHD requires a multidisciplinary approach [125]. For the purpose of timely diagnosis and early initiation of therapy and to prevent the development of life-threatening conditions and disabilities, it is necessary to conduct a systematic, thorough assessment of patients’ organs and systems. This also helps in evaluating the response to therapy and in determining further treatment strategies. Currently, along with the development of pharmacology, new drugs and treatment regimens for GVHD are appearing on the market; it is in the development of new approaches to the treatment of this life-threatening condition that there is hope for a reduction in complications and mortality after allogeneic HSCT.

## Figures and Tables

**Figure 1 pharmaceuticals-15-01100-f001:**
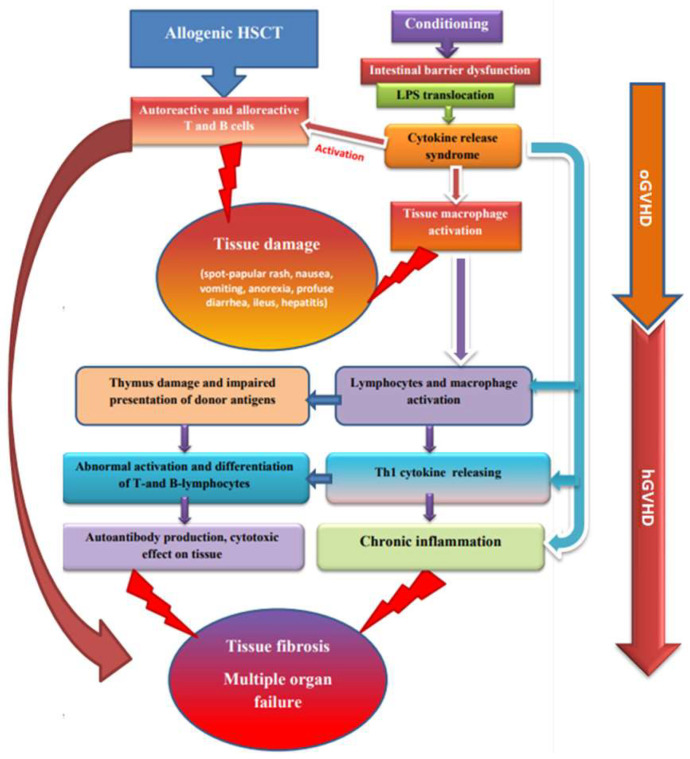
Relationship between aGVHD and cGVHD pathogenesis.

**Figure 2 pharmaceuticals-15-01100-f002:**
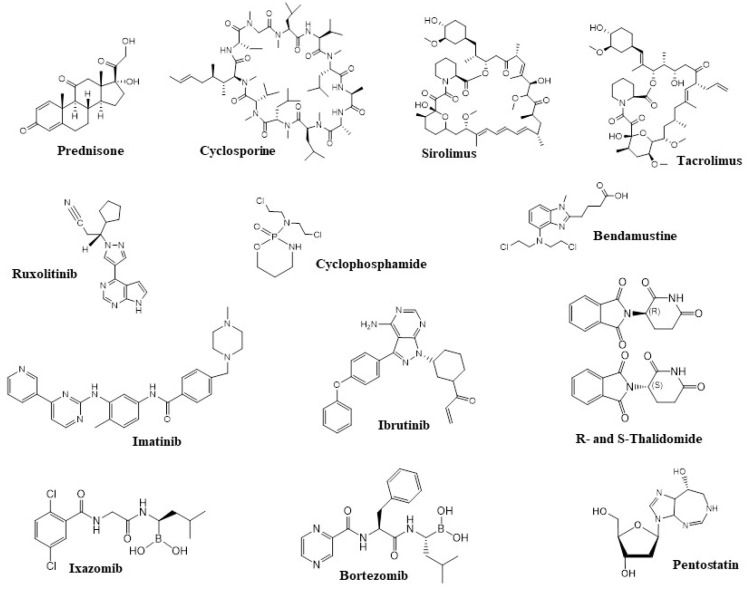
Structures of active components of the drugs currently used for the treatment of GVHD or in drug candidates under development.

**Figure 3 pharmaceuticals-15-01100-f003:**
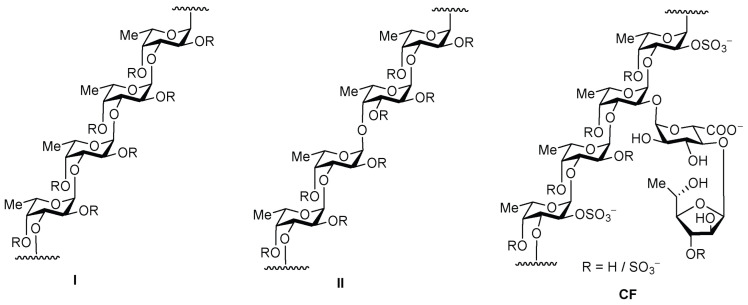
Two types of homofucose backbone chains in brown seaweed fucoidans (**R** depicts the places of potential attachment of carbohydrate substituents, sulfate, and acetyl groups) and the structure of fucoidan **CF** from *C. flagelliformis* seaweed.

**Figure 4 pharmaceuticals-15-01100-f004:**
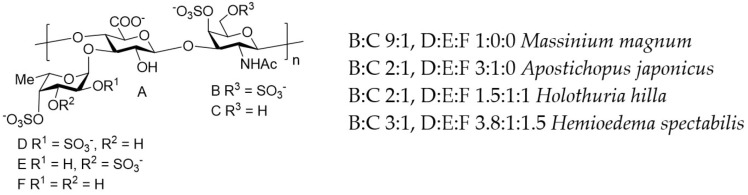
The main structural features of fucosylated chondroitin sulfates from the sea cucumbers *Massinium magnum*, *Apostichopus japonicus*, *Holothuria hilla*, and *Hemioedema spectabilis*, as examples of natural polysaccharides of this type.

## Data Availability

Data sharing not applicable.

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
