# Peer review of "Novel and Promising Strategies for Therapy of Post-Transplant Chronic GVHD"

_pharmaceuticals, 2022, doi:10.3390/ph15091100_

Round 1

Reviewer 1 Report (Previous Reviewer 2)

The authors have improved their manuscript.

Author Response

Responses to reviewers

First submitted as pharmaceuticals–1863200

Title: Novel and promising strategies for therapy of post-transplant chronic GVHD

Responses to Reviewer 1

Reviewer 1: Comments and Suggestions for Authors

The authors have improved their manuscript.

 Authors: We thank Reviewer 1 for the consideration of the manuscript and positive note. Present version was subjected to editing of English language and style as recommended.

Reviewer 2 Report (Previous Reviewer 3)

The authors have made a good attempt to respond to previous reviewers' concerns. 

Comments:

Lines 180-190--Ruxolitinib is now approved in US for chronic GVHD. 

Under section 3, would make sub-headings for ECP, TReg manipulationCAR-Tregs, MSCs, etc. 

Likewise under section 4, would create subheadings as one agent's description runs into another's. 

The description of Rho inhibitors seems incomplete and inaccurate. Belumosudil and not bendamustine should be discussed.  This needs to include discussion of Th17 effects as well.  Bendamustine is not used in GVHD therapy to my knowledge other than as a replacement for Post-cyclophosphamide.  In line 440-, what is BEX? . 

Would shorten the section on hematopoietic dysfunction and hemostasis effects and eliminate most of the section on fucoidans. This needs to be better tied to the rest of the review. 

Line 140--?locking should be blocking

Line 456--CSF-1 is not KGF; see last review. 

Please check all references; for example; reference one Bazar is Blazar. 

Author Response

Responses to reviewers

First submitted as pharmaceuticals–1863200

Title: Novel and promising strategies for therapy of post-transplant chronic GVHD

Responses to Reviewer 2

We thank Reviewer 2 for the consideration of the manuscript and the notes. Please find our answers below.

Reviewer 2: Lines 180-190-Ruxolitinib is now approved in US for chronic GVHD. 

Authors: The phrase «Ruxolitinib in 2021 approved by FDA for the cGVHD [26]» was added to the text.

Reviewer 2 Under section 3, would make sub-headings for ECP, TReg manipulationCAR-Tregs, MSCs, etc. 

Authors: Subheadings have been created.

Reviewer 2 Likewise under section 4, would create subheadings as one agent's description runs into another's. 

Authors: Subheadings have been created.

Reviewer 2 The description of Rho inhibitors seems incomplete and inaccurate. Belumosudil and not bendamustine should be discussed.  This needs to include discussion of Th17 effects as well. 

Bendamustine is not used in GVHD therapy to my knowledge other than as a replacement for Post-cyclophosphamide. 

Authors: New materials on promising strategies for cGVHD therapy with Belumosudil are now reviewed.

Reviewer 2 In line 440-, what is BEX? 

Authors: “BEX” was corrected to make “BEN”.

Reviewer 2 Would shorten the section on hematopoietic dysfunction and hemostasis effects and eliminate most of the section on fucoidans. This needs to be better tied to the rest of the review. 

Authors: This section was reduced, in particular one duplicated place was deleted.

 Reviewer 2: Line 140--?locking should be blocking

Authors: this misprint was corrected.

Reviewer 2 Line 456--CSF-1 is not KGF; see last review. 

Authors: “KGF” was substituted with “CSF-1 (colony-stimulating factor 1)”.

Reviewer 2 Please check all references; for example; reference one Bazar is Blazar. 

Authors: References were re-checked, above misprint was corrected.

This manuscript is a resubmission of an earlier submission. The following is a list of the peer review reports and author responses from that submission.

Round 1

Reviewer 1 Report

In the current article Kostareva et al. describe novel pharmaceutical therapies for chronic GVHD. However, the various treatment forms of cGVHD are inadequately described and although data on specific treatment are scares, a detailed and precise description is required. E.g. The recently published data on Rho inhibitor (Belumosudil), newly developed JAK2 inhibitors (Baricitinib), new CD20 antibodies +++ are not presented, Previously meta-analysis on the efficacy of various GVHD treatment and perspectives on cost analysis is not included (that shows that MTX is far superior compared with other therapies from a cost/efficacy perspective, PMID: 29550629). Data from randomized controlled trails (e.g. data on MMF) demonstrating that some treatments might have a very low activity in cGVHD, is not presented

In its current form, the review is neither detailed nor precise and should be rejected

Reviewer 2 Report

The authors have submitted a short but potentially interesting review on novel pharmaceutical therapies for chronic graft-versus-host disease (cGVHD) in the setting of hematopoietic stem cell transplantation.

While the review is potentially interesting, it currently does not offer a substantial contribution to the literature, for example when compared to Hamilton BK, Updates in chronic graft-versus-host disease, Hematology Am Soc Hematol Educ Program 2021 (1): 648–654.

To publish this review in a journal like Pharmaceuticals, the authors would need to significantly expand sections 7 and 8 by at least 3-4 fold with additional background information on the discussed agents, its molecular and cellular mechanism of action and more in-depth information from the discussed clinical studies.

In addition, Figure 1 is titled "Relationship between aGVHD and cGVHD pathogenesis", but this figure does not show the relationship between aGVHD and cGVHD because these parts of the figure are not explicitly connected with each other.

Reviewer 3 Report

Comments:

1.  Many in the trasnplant community would not call it a "radical" therapy as it has been accepted since the 1980's as standard of care in some diseases. 

2.  In line 49, what is meant by "sickness?

3.  In section 5, how should those patients who are already on calcineurin inhibitors be approached?  

4.  LIne 153; would describe some ibrutininb side effects like muscle cramps which may make it difficult to administer in GVHD patients with musculoskeletal effects. Ibrutinib is then described again in line 168 to 182; would consolidate this. 

5,  For Table 1, would include references for each agent. 

6.  CSF-1 is not a keartinoycyte growth factor. 

7  In the section on new agents, etanercept, pentostatin, etc have been examined for some time, so would not characterize them as new or experimental.  Would review clinical trials websites and report on new Jak inhibitors and other agents that are being examined.  It is fine to mention the older ones, but most of them are no longer used regularly. 

8 LIne 255--?pro should be anti-inflammatory

Minor: 

Would define aHSCT when first used. 

Figure 1 references reference 3.  I this adapted from that or borrowed from that? 

Would define RORyt. 

Line 264--would define hRTPH